# Diversity and Functional Relevance of Canopy Arthropods in Central Europe

Andreas Floren [1,2,*], Karl Eduard Linsenmair [2] and Tobias Müller [1]

1   Department of Bioinformatics, Biocenter, University of Würzburg, Am Hubland,
    D-97074 Würzburg, Germany
2   Department of Animal Ecology and Tropical Biology, Biocenter, University of Würzburg, Hans Martin-Weg 5,
    D-97074 Würzburg, Germany
*   Correspondence: floren@biozentrum.uni-wuerzburg.de

**Abstract:** Although much is known about the ecology and functional importance of canopy arthropods in temperate forests, few studies have tried to assess the overall diversity and investigate the composition and dynamics of tree-specific communities. This has impeded a deeper understanding of the functioning of forests, and of how to maintain system services. Here, we present the first comprehensive data of whole arthropod communities, collected by insecticidal knockdown (fogging) from 1159 trees in 18 study areas in Central Europe during the last 25 years. The data includes 3,253,591 arthropods from 32 taxa (order, suborder, family) collected on 24 tree species from 18 genera. Fogging collects free-living, ectophytic arthropods in approximately the same number as they occur in the trees. To our knowledge, these are the most comprehensive data available today on the taxonomic composition of arboreal fauna. Assigning all arthropods to their feeding guild provided a proxy of their functional importance. The data showed that the canopy communities were regularly structured, with a clear dominance hierarchy comprised of eight 'major taxa' that represented 87% of all arthropods. Despite significant differences in the proportions of taxa on deciduous and coniferous trees, the composition of the guilds was very similar. The individual tree genera, on the other hand, showed significant differences in guild composition, especially when different study areas and years were compared, whereas tree-specific traits, such as tree height, girth in breast height or leaf cover, explained little of the overall variance. On the ordinal level, guild composition also differed significantly between managed and primary forests, with a simultaneous low within-group variability, indicating that management is a key factor determining the distribution of biodiversity and guild composition.

**Keywords:** temperate forests; insecticidal knockdown; community structure; functional diversity; guild constancy; forest management; pristine forests; Bialowieza





## 1. Introduction

Forty years after the discovery of high biodiversity in tropical rain forest trees [1], canopy research has also become established in temperate Central Europe [2–4]. Nevertheless, our knowledge of the diversity and frequency distribution of arthropods in tree crowns is still limited. This can be seen, for example, in the remarkably high numbers of red-listed beetle species collected from floodplain forest trees in Saxony–Anhalt (Germany), which accounted for 23% of all beetle species and 12% of all specimens [5]. Long-term studies which record the spatio–temporal dynamics of canopy communities comprising all taxa have not yet been performed. Such research, however, is a precondition for a deeper understanding of how canopy arthropods are involved in the maintenance of ecosystem functioning and services [6,7].

The research presented here considered arthropods living freely in the canopy. For this ectophytic fauna, a good approximation of its diversity and distribution can be obtained by

using insecticidal knockdown (fogging), which captures arthropods in approximately the number in which they occur in the trees [8]. Based on the fogging data, specimens can be assigned to their feeding guild, to assess their function [4,9] (see also Table S3 of this study). Such an approach had not yet been attempted, even though the first fogging studies were conducted decades ago [10]. Guild analyses based on these data focused on the number of species per feeding guild [11,12], but did not raise the question of the functional relevance of the canopy fauna. Unlike Erwin's study on beetle diversity in tropical rain forest trees [1], which triggered a paradigm shift in biodiversity research, the ecological significance of canopy fauna had not yet been considered in a broader ecological context. An attempt to functionally characterize whole communities was therefore overdue.

Currently, there is little information on the distribution of guilds in the trees. The situation is further complicated by the fact that authors use different guild definitions, and often restrict analyses to a few taxa [4,9,13–16]. This still prevents an overall view, leaving unanswered the question of the mechanisms maintaining the functional importance of the canopy fauna. Our data provide the basis for closing these knowledge gaps for temperate forests in Central Europe.

In this work, we relied on the most comprehensive dataset available, to examine how arthropod communities of temperate forest trees are composed, and whether composition follows a regular pattern. Which groups dominate in the canopy, which occur sporadically, and how stable is the composition? Based on the quantitative fogging data, we then asked how the guild composition of different tree species differed from one other. Firstly, we compared deciduous and coniferous trees, which were expected to deviate greatly in taxa composition due to pronounced differences in leaf morphology, crown structure and phytochemistry. We analysed, in detail, the guild composition of *Quercus* trees, which harbour a highly diverse fauna [17,18], and which were sampled in the greatest number. We examined how study area, year and tree-specific factors—tree height, girth in breast height and leaf cover—influenced guild composition.

Forest management strongly impacts tree diversity, the age-composition of trees and, most importantly, crown structure, thereby determining forest climate and the distribution of biodiversity [19]. We therefore included management in the analysis. As there are hardly any pristine forests left in Central Europe, the effects of forest management have been investigated relative to different management practise or, at best, to unmanaged forests [20]. We studied this contrast, by comparing commercial forests with pristine forest sanctuaries in the Bialowieza Forest in eastern Poland. These forests are considered to be the most natural primary forest remnants in Central Europe [21–23], and were therefore used to evaluate the management effects.

## 2. Materials and Methods

Canopy arthropods were sampled over the course of 25 years (1995–2020) in different countries and sites in Central Europe. These were, in descending order of foggings: Germany (805), Poland (275), Belarus (40), Slovenia (19) and Romania (13) (Figure 1). In total, 1152 foggings were carried out, covering 18 tree genera and 26 species. The number of foggings per tree genera varied between 380, for *Quercus*, and single foggings on *Malus sylvestris* and *Pyrus pyrestris*. Field work was carried out in different forest types, which are characterised in greater detail in (Table S1).

Canopy fogging is a highly effective method of collecting ectophytic, free-living arboreal arthropods in a quantitative and tree-specific way [5,8]. Fogging is relatively easy to apply, and takes only a few minutes in the field. The impact on the ecosystem is greatly reduced by using natural pyrethrum at a concentration of less than 1%. The insecticide is diluted in a highly purified white oil with no chemical additives. Natural pyrethrum is highly specific to arthropods and causes uncoordinated movements, so that the arthropods eventually fall from the leaf or branch surfaces down into the collecting trays [8]. It is destroyed in direct sunlight within hours, and does not leave any harmful substances in the trees. This contrasts with synthetic pyrethroids and other insecticides, which make a much

greater impact, due to their long-lasting persistence in the environment [24,25]. Depending on the local weather conditions, the effect of the insecticide is limited to a narrow radius of about 50 m around the examined tree, in the direction in which the fog drifts. As a result of this careful procedure, most of the knocked-down arthropods recover unless they are not collected. Collection sheets were installed beneath each tree, covering at least 80% of the crown projection study area. By exact positioning of the collecting sheets, arthropods from neighbouring trees were excluded from the sample. Foggings were carried out early in the morning or in the evening, when there was little air movement. Fogging was not performed when leaves were moistened by dew, which limits arthropod mobility. All the arthropods that had dropped onto the collecting sheets, two hours following fogging, were collected and conserved in 80% ethanol. Most samples were collected in June, before the juveniles had pupated, and when diversity was at its highest.

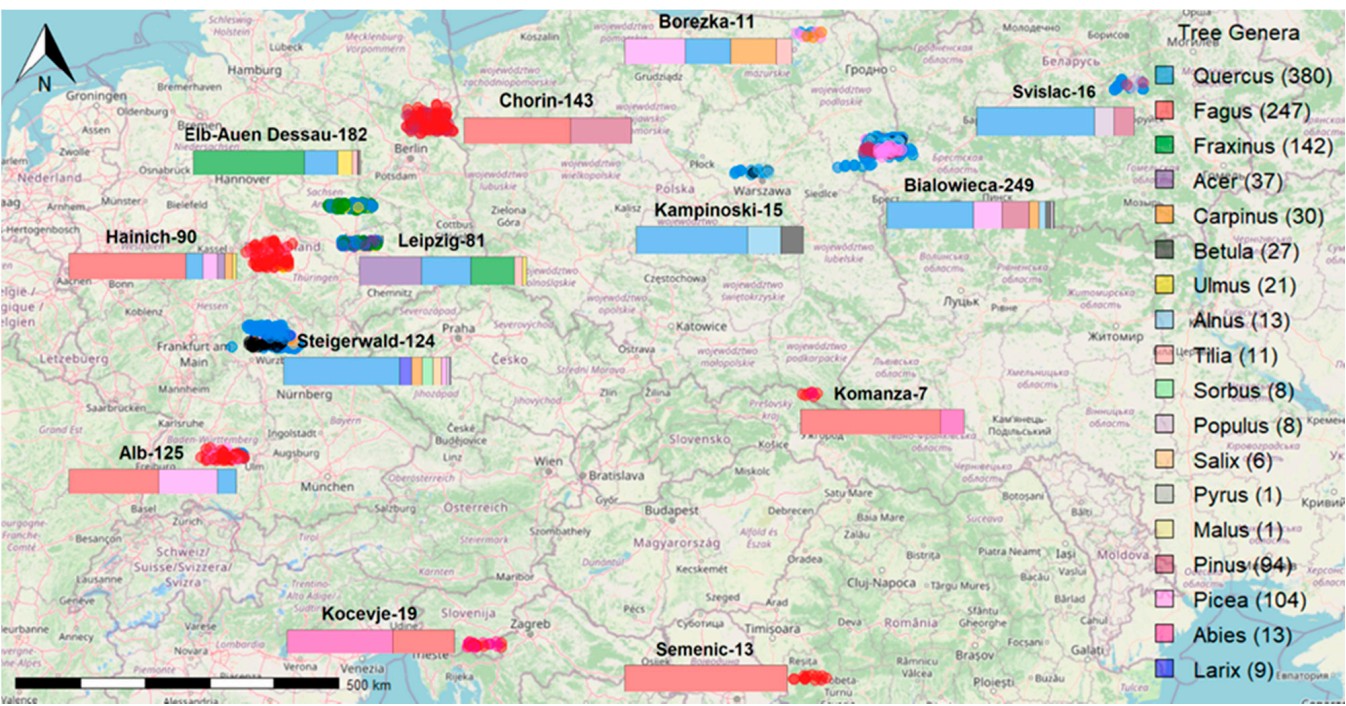

**Figure 1.** The map of study sites in Central Europe shows an almost representative coverage. Trees fogged were plotted and visualised as differently coloured dots, which were jittered for better visualisation. The coloured bar graphs show the percentage of tree genera fogged per site, together with the total number of foggings in brackets. All sites with seven trees, or more, are shown.

The high accuracy of fogging, the low ecosystem impact, and the unselective collection of all arthropod taxa which move freely in the canopy, underlines that the criticism of fogging as a destructive mass collection method is not justified—particularly in comparison to other quantitative collection methods, such as flight intercept traps or malaise traps. [26]. However, it is important to mention that the fogging procedure must be performed by experienced persons, as precisely as possible, to ensure low disturbance and highly reproducible results.

All arthropods were sorted to the ordinal, subordinal or family level (Table S2). In accordance with their numerical dominance and ecological importance, Formicidae (Hymenoptera) were treated separately, as were Aphidoidea (Hemiptera). Within the order Hemiptera, we distinguished suborders Heteroptera (true bugs) from Homoptera (plant suckers). Numbers of Thysanoptera, Acarina and Collembola varied greatly from tree to tree [27], preventing a meaningful assessment of their true frequencies; therefore, they were not considered in the analysis.

All arthropods of each community were partitioned into discrete feeding guilds, to provide a functional link between the communities and their overall impact (Table S3). We were aware that such an approach could be complicated by ontogenetic niche shifts, which describe changes in the diet and habitat use of juveniles and adults. However, since we were only referring to communities collected in June, we considered only the corresponding developmental stages of all species for the guild classification. The few single adult Lepidoptera were not included separately in the analyses, and all caterpillars were assigned as leaf-chewers. For most taxa, assignment to a feeding guild was unambiguous, but for Coleoptera and Heteroptera, which comprise different feeding guilds, assignment was based on species identifications done by specialists. Due to the high time, financial and taxonomic effort this was not possible for Diptera; consequently, this ecologically important, species-rich and functionally diverse group could not be taken into account in the guild analyses.

The following feeding guilds were distinguished: phytophages (separated into chewers and sap-suckers); zoo-phytophages (zoo.phy)—mainly heteropterans; zoophages (zoo); parasitoids (para); xylophages (xyl); saprophages (sap); mycetophages (myc); and epiphyll-grazers (grazer). The only ant species collected in large numbers was *Formica polyctena* (Formicinae), which represented 79.1% or 110,915 specimens of all ants collected. This predacious species enters the trees from large ground nests, to forage in the canopy. As all other ant species were collected in low numbers per tree, ants were considered zoophages. The largest proportion of Hymenoptera were parasitoids, and only a few phytophagous larvae of sawflies were collected. Xylophages and mycetophages were mainly represented by beetles. Only four species of Aradidae (Heteroptera), living under the bark of trees, were collected in a few specimens.

We described the overall distribution of arthropod abundance per taxon in the trees. Then, we evaluated the importance of the tree genera—where tree species of the same genus were grouped together (mainly *Quercus robur* and *Q. petraea*)—year and study site, as ecological drivers for guild composition.

Thus, we dealt with the following overall nested sampling: selected data level 1: taxonomic abundance composition of all individuals collected from 1152 trees fogged in Central Europe; June (N = 949), August (N = 100) and September (N = 101):

- Comparison between tree genera (violin plots, bar plots of taxa ranks);
- Comparison between deciduous and coniferous trees (boxplots of individuals per tree genera).

Selected data level 2: for a dataset of (N = 319) trees fogged in June, the whole guild composition was determined:

- Comparison of guild composition between tree genera (level-plot of individual percentage per guild, boxplot of deciduous vs. coniferous tree genera). To verify the significance of the results, the analyses were then performed on a more homogeneous, smaller, nested data set. For this purpose, we chose the oaks in Poland and, in particular, in the Bialowieza Forest, which also enabled comparison between primary and commercial forests located in the same forest matrix. Selected data level 3: *Quercus* trees in Poland (N = 103):
- Guild composition (PERMANOVA; Permutational Multivariate Analysis of Variance using Distance Matrices); adonis2 function, as implemented in vegan R package [28].

Selected data level 4: *Quercus* genera in the Bialowieza Forest in 2001 and 2002 (N = 77):

- Guild composition in primary and managed forest sites (logistic regression).

*Bialowieza Forest*

If not stated otherwise, we used only trees fogged in June for guild analysis. Species that achieved exceptionally high abundance under certain local environmental conditions likely to impact the whole community, and rare species, were identified through exploratory analyses. Further details are given in the respective paragraphs. Statistical analyses

were performed using the software R version 4.0.2 and the packages of the Bioconductor project [29]. The map of the study area was produced by using R packages rosm [30] and prettymapr [31]. Communities of arthropod individuals and compositional data were visualized by boxplots and tested by rank-based Kruskal–Wallis tests. As a post hoc test, we applied the pairwise Wilcoxon rank test with multiple testing correction according to Benjamini–Hochberg (BH) [32]. We assumed *p*-values to be weakly significant if $p < 0.05$ (*), significant if $p < 0.01$ (**) and strongly significant if $p < 0.001$ (***).

The distributions of all arthropods (selected data level 1) were visualised as violin plots [33] with overlaid boxplots. To visualise frequency distribution of taxa, we plotted the mean rank of each taxon per tree genera with standard errors.

For each tree-specific fogging sample in June, we derived eight relative proportions of guilds for all taxa (selected data level 2) (see functional classification paragraph). The pairwise difference of guild compositions of two trees, i and j, was calculated by an all-against-all distance matrix based on the Euclidean distance $(D)_{ij}$. These guild differences were modelled by the adonis2 function [28,34]: D ~ Year + Tree Genus + Management + Leaf Cover + Tree Height + Area + Altitude, where Year, Tree Genus, Management and Area were modelled as categorial factors, and Leaf Cover, Tree Height and Altitude as numerical factors. Leaf cover was measured for each tree crown as a relative proportion against the sky. Management was a categorial binary factor with two levels, "Primary Forest" (no management) and "Disturbed Forest" representing age-class forest sites from 10 to 80 years of *Quercus* embedded within the Bialowieza Forest matrix.

The significance of the factors was derived by a permutation-based statistical test, using the adonis2 function of the vegan package [28], based on 99,999 permutations. The same model and analysis were applied to all combined deciduous trees and conifers. To model the guild differences more precisely, we reduced the variability of data by focussing on *Quercus* trees in Poland fogged in the study areas Bialowieza, Borezka, Kampinoski and Nurzec (selected data level 3). We focused on *Quercus* in Bialowieza for the years 2001 and 2002, with annotated forest sites within the Bialowieza Forest matrix (selected data level 4). For these small sampling regions, we directly used tree GPS coordinates (WestEast, NorthSouth) in the model, instead of the categorial factor area, to model spatial effects in the data. Guild composition on *Quercus* trees in the Bialowieza Forest was analysed by logistic regression. Due to overdispersion, we used logistic regression based on quasibinomial family: Proportion of each individual Guild(i) ~ Year + Management + Leaf Cover + WestEast + NorthSouth. Year had two categorial levels: 2001, with the reference level 2002. Finally, we applied to all models a backward model selection, based on the F approximation in the analyses of deviance. We modelled guild composition in Bialowieza, in terms of dependence of Year, Management, WestEast, NorthSouth and LeafCover, by a logistic regression. Since AIC or BIC were not available for such models, we conducted model tests by applying the stepwise F-test, as implemented in the drop1 R function.

## 3. Results

### 3.1. Arthropods in Trees

This analysis was based on the total dataset covering 1152 foggings and 3,699,136 arthropods. Thysanoptera (271,359), Acarina (109,557) and Collembola (71,701) were excluded from the analyses, because their numbers varied greatly between trees, and could not be sampled representatively by fogging. This reduced the data to 3,246,519 specimens. The tree genera fogged in highest numbers were *Quercus* (380), *Fagus* (247), *Fraxinus* (142) and *Picea* (104). The most frequent taxa were Diptera (586,037 specimens or 18.1% of the total), Homoptera (499,463 specimens; 15.4%), Coleoptera (358,202; 11.0%), Heteroptera (354,998; 10.9%) and Psocoptera (347,474; 10.7%). Counts and percentages of all taxa are summarised in Table S2.

The taxa were not randomly distributed among the trees, but were arranged in a specific order (Figure 2). Based on our comprehensive dataset, we were able to assign the taxa to three categories, according to their frequency distribution: "**major taxa**" com-

prised all those groups that were regularly collected from all fogged trees, representing, on average, a minimum 5% of all individuals in canopy communities. Exceptions were Lepidoptera (3%) and Arachnida (2%), which were also considered "major taxa" because they were collected from every tree. "Major taxa" represented the largest proportion, namely 2,730,874 individuals or 84.12%, which included the following taxa in descending order: Diptera, Homoptera, Hymenoptera, Heteroptera, Coleoptera, Psocoptera, Lepidoptera and Arachnida (mainly Araneae, Table S2). The "major taxa" themselves differed in frequency distribution, distinguishing dominant Diptera and Homoptera and the less frequent Lepidoptera and Arachnida from the remaining taxa (Figure 3A). The second group were the "**minor taxa**", which occurred on most trees, but in significantly lower numbers, providing, on average, less than 1% of all specimens in a community. These were the Aphidae (Homoptera, Hemiptera), Formicidae (Hymenoptera), Neuroptera, Blattodea, Orthoptera and Dermaptera. Altogether, they represented 421,954 individuals or 13%. Aphidae and Formicidae did not occur regularly on all trees, and varied greatly in numbers. They could also become locally dominant. The last category comprised taxa that were collected only "**sporadically**", and usually in low numbers. These were Plecoptera, Trichoptera, Ephemeroptera (which could become numerically dominant after mass hatching) and Mecoptera, Raphidioptera, Megaloptera, Odonata, Isopoda, Pseudoscorpiones, Archaeognatha, Chilopoda and Diplopoda, including also rare Mollusca and Annelida. Altogether, they comprised 93,691 individuals or 2.89% of the total.

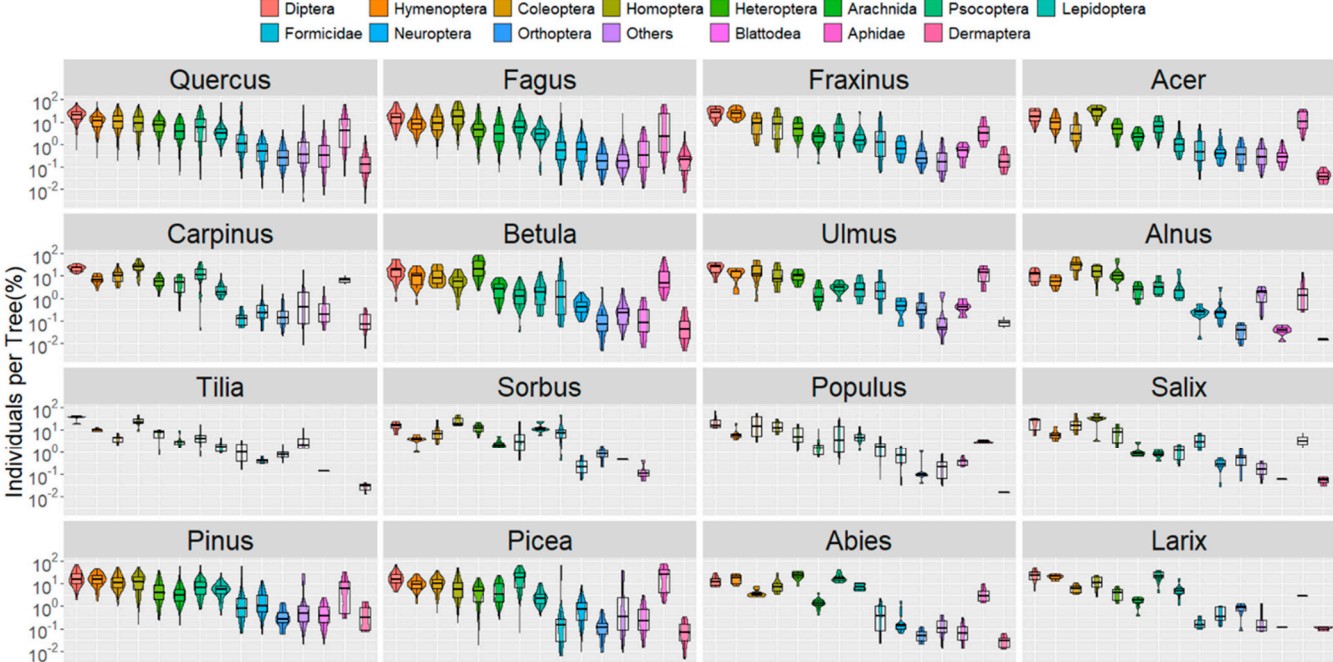

**Figure 2.** Violin plots with overlayed boxplots show how much arthropod taxa contribute to the canopy communities in the tree genera investigated. Compact violins indicate low variance in numbers, while extended violins show high betweensample variability. Tree genera are sorted according to the number of foggings; taxa are sorted according to the median. *Quercus* and *Pinus* were used as a reference for deciduous and coniferous trees, respectively. Two trees fogged in only one individual (*Malus* and *Pyrus*) are not displayed.

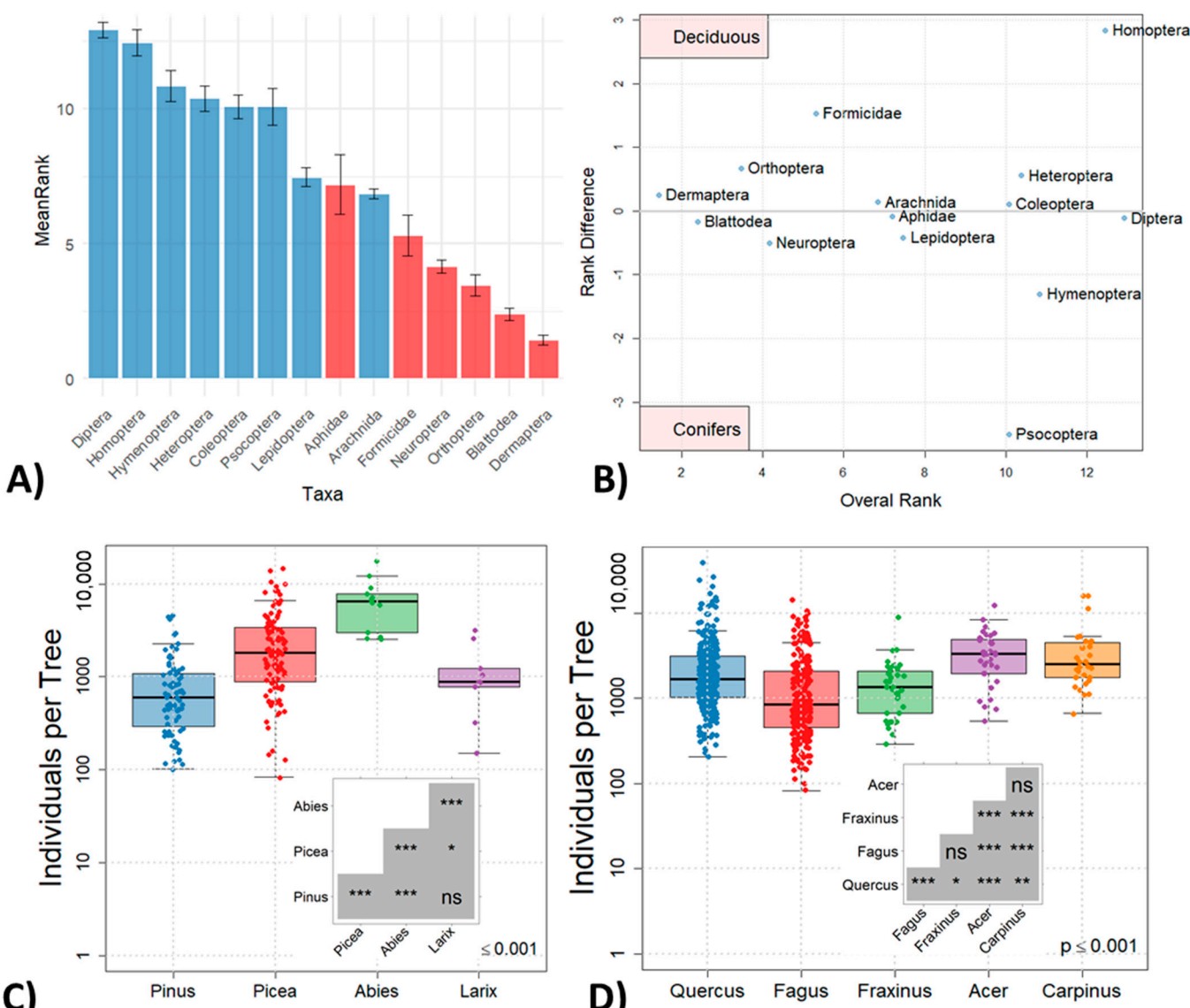

**Figure 3.** (**A**) Arthropod taxa were arranged in a distinct rankorder on the trees, as visualised by mean ranks with standard errors (+/− 1 standard error of the mean), with the highest rank equal to 14. "Major taxa" are in blue; "minor taxa" are in red (see text) (**B**). Plotting the rank differences of taxa on deciduous and coniferous trees emphasises taxa preferences. The further to the right a taxon is on the *x*-axis, the more often it was found in a high rank; the further up or down on the *y*-axis, the more taxa were preferentially collected from deciduous or coniferous trees. (**C**,**D**) show the overall distribution of arthropods on conifers and deciduous trees; both indicate overall differences within deciduous trees and conifers (Kruskal Test, $p < 0.001$). The difference in the means of all tree genera was compared, and the associated adjusted significance levels are displayed in the inlays (Wilcoxon test with multiple testing correction), $p < 0.05$ (*), $p < 0.01$ (**), $p < 0.001$ (***).

The regular taxa arrangement was often interrupted by occasional mass appearances of individual species, resulting in significant differences between communities. Examples were Homoptera, which were more frequent on *Acer* and *Carpinus*, while the number of aphids varied greatly between trees, and could be very high on some trees, while being absent on neighbouring conspecific trees. True bugs dominated on *Betula*, mainly represented by the species *Kleidocerys resedae* (Panzer, 1797, Lygaeidae), whilst parasitic Hymenoptera were collected in the highest proportions from *Fraxinus* trees. In the Bialowieza Forest, the stonefly *Nemoura cinerea* (Retzius, 1783) dominated trees indiscriminately after hatching

synchronously. In contrast, caterpillar calamities, such as occurred in the Steigerwald in 1996, resulted in complete defoliation of all oaks; except for the trees fogged. Another example that belongs in this category is the dominance of predacious *Formica polyctena* ants (Förster 1850) in Münnerstadt, which enter trees from huge ground nests to forage in the canopy. Such occasional mass appearances represent special ecological conditions, and must be studied separately in terms of their functional significance.

In the following, we describe in detail the differences in community composition between deciduous and coniferous trees. Our analyses comprised 902 fogged trees in June, and 2,435,677 arthropods. Significant differences in the rank order were observed between deciduous trees (N = 697 foggings and 528,511 arthropods) and conifers (N = 205 foggings and 192,516 arthropods; Wilcoxon test $p$ = 0.0223). Plotting the rank differences of taxa against their maximum rank (Figure 3B) showed that Psocoptera and parasitic Hymenoptera were preferentially collected from conifers, while Homoptera and Formicidae, Heteroptera, Orthoptera and Dermaptera were, on average, more frequent on deciduous trees. Ranking of taxa is a characteristic feature between different trees, but ranking does not indicate absolute frequency values. This is illustrated in Figure S3, which shows multiple testing-adjusted significance levels of the mean abundance of taxa between coniferous and deciduous trees. The mean numbers of Arachnida, Aphids, Coleoptera, Blattodea, Lepidoptera (mainly caterpillars), Neuroptera and parasitic Hymenoptera did not differ significantly between coniferous and deciduous trees. The differences among conifers and deciduous trees were also evident in the number of arthropods. The fewest specimens were collected from *Pinus* (median 603) and *Larix* (median 892), due to their small crown size and low leaf cover. Most arthropods were sampled from *Abies* (median 6555), which differed significantly from *Picea* (median 1943; Figure 3C). *Acer* (median 3653) and *Carpinus* (median 2579) harboured the most abundant communities, followed by *Quercus* (median 1783) and *Fraxinus* (median 1548). The arthropod numbers were lowest on the *Fagus* trees (median 862; Figure 3D).

## 3.2. Functional Classification of Canopy Communities

Starting from the general classification based on individuals, we next characterised communities based on their guild composition. For this purpose, we used 319 foggings that were fully annotated to feeding guilds. Only tree genera from which at least 10 trees were fogged were included. In total, these foggings comprised 682,904 arthropods. First, we compared deciduous trees (256 trees; 563,608 specimens) and conifers (63 trees; 119,296 specimens) which differed only in the proportion of grazers (Wilcox test $p$ = 0.001). Phytophages dominated on both deciduous trees and conifers, followed by parasitoids (Hymenoptera), grazers and zoophages (Figure 4A). When comparing each tree genus in terms of guild composition, we found only a few significant differences (Figure 4B). Grazers were significantly more abundant on *Picea* than on *Acer*, *Fraxinus*, *Quercus* and *Fagus* (Figure S2), while the proportion of xylophages was higher on *Picea* than on *Quercus* and *Fagus*. As for mycetophages, *Fagus* and *Quercus* differed only from *Fraxinus* and *Acer*, while the proportion of zoophages was 7.3 times greater on *Quercus* than on *Acer*.

We used multivariable adonis analysis [28] for modelling the Euclidean distance of the guild composition between trees, and found highly significant differences between tree genera, the study area and the year of the investigation. Based on the full dataset, the binary factor Management was weakly significant in explaining 0.4% of the total variance of the given distance matrix (Figure 5). Tree height, leaf cover and altitude were not significant. The overall proportion of explained variance was 55% (Table S4). The same approach was used for conifers and deciduous trees: here, the study area explained most of the total variance: 22.5% and 32.7%, respectively. For conifers alone, Year and Management were not significant, but tree height was weakly significant. The overall explained variance was 62% for deciduous trees and 56% for conifers.

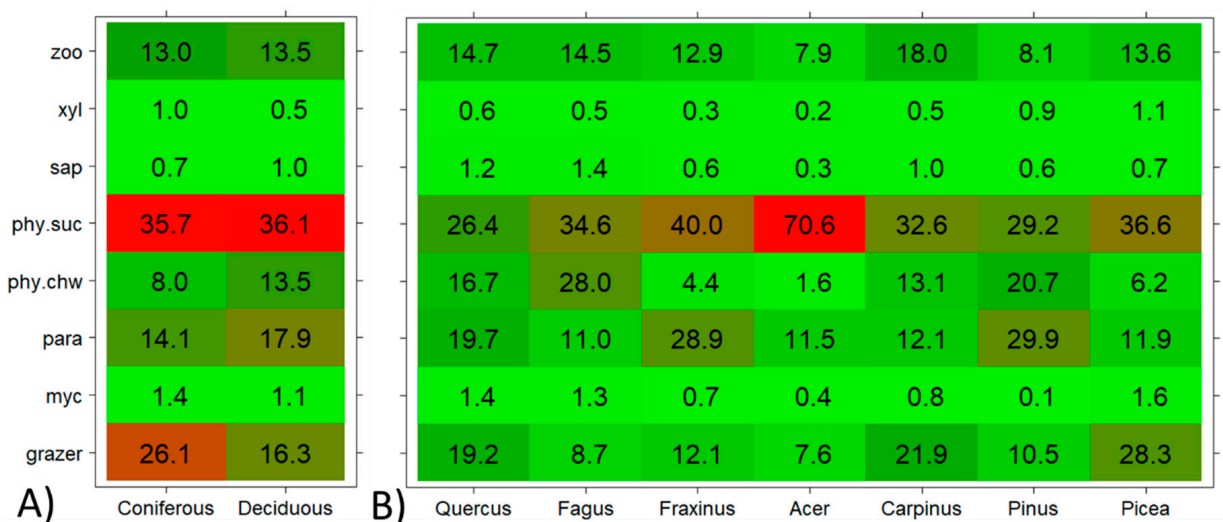

**Figure 4.** Relative proportions of beetles per feeding guild on deciduous and coniferous trees (*Picea* and *Pinus* (**A**)) and on tree genera (**B**). The darker the shade of green, the greater the proportion of the particular guild; the highest values are shown in bright red. We used only tree genera with at least 10 fogged trees in the analyses. The total number of foggings was 319.

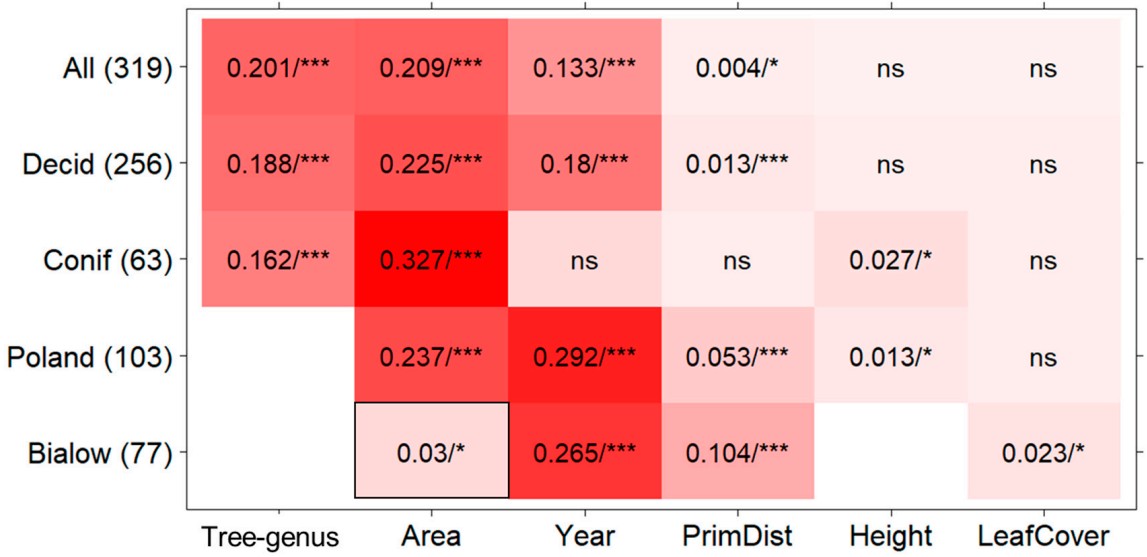

**Figure 5.** Results of the (adonis) regression models of guild composition, showing $R^2$ values for significant coefficients based on different study strata (*y*-axis), with the number of fogged trees in brackets. On the *x*-axis, the $R^2$ with the associated significance level is shown. Dropped factors are indicated by white boxes, while non-significant factors are depicted with "ns". For the condensed Bialowieza model (Bialow) only, we used the coordinates of each sampled tree as numerical factors in the model, in which only NorthSouth was weakly significant ($R^2 = 0.03$ framed box for Area). The factor Height (tree height) was dropped in this model, due to strong collinearity with the factor Management. $p < 0.05$ (*), $p < 0.001$ (***).

In the following, we successively reduced the variability in the data by focussing only on *Quercus* in Poland (samples in 2001: 33 oaks in Bialowieza; samples in 2002: 44 oaks in Bialowieza and 3 oaks in Borezka; samples in 2003: 8 oaks in Bialowieza, 10 oaks in the Kampinoski Park and 5 oaks in the Nurzec forest). The results were consistent with the previous outcomes, in recognising the highly significant influence of the factors Year ($R^2 = 29.2\%$), Area ($R^2 = 23.7\%$) and Management ($R^2 = 5.3\%$). This model explained

60.5% of the total variance. Finally, we focused on guild composition on *Quercus* trees in Bialowieza. In primary sites, 30 oaks were compared with 47 oak trees in managed sites. The 77 foggings comprised, in total, 251,891 arthropods. For this most condensed model only, we used the coordinates of each tree instead of Area as numerical factor. In this set-up, Year explained most of the variation ($R^2$ = 26.5%), followed by Management ($R^2$ = 10.4%); NorthSouth (framed rectangle) and LeafCover were also significant, but explained little of the variance. This model explained 43.2% of the total variance.

These analyses show that the functional composition of canopy communities is primarily determined by tree-genus, study area, year, and forest management, while tree-specific factors, such as tree height and leaf-cover, play only a minor role.

We studied the effects of forest management on guild composition, using oaks in primary sites, and those in plantations in the Bialowieza Forest, in 2001 and 2002. In the primary sites, the samples represented 218,512 arthropod specimens, and in the managed sites 33,379 specimens. Again, the factor Year was a significant influence, with grazers and sap-suckers collected in lower proportions in 2002, while proportions of all other guilds had increased (Table 1). We also found that proportions of zoophages were lower in the primary forests compared to the managed sites. Although the effect of size was large for other guilds, e.g., for grazers and sap suckers, the differences were not significant, due to high variability between trees. Parasitic Hymenoptera decreased with increasing leaf-cover, while saprophages and zoophages increased. Adjusting the data to geographical coordinates, we also observed weakly significant effects for plant-suckers and plant-chewers, and a significant association for saprophagous beetles (not shown).

**Table 1.** Results of modelling guild composition by logistic regression of 77 *Quercus* trees in Bialowieza in 2001 and 2002, in primary and managed forests, adjusted for leaf cover. Variables deleted by model test are displayed by white boxes. Numbers indicate odds ratios of the associated coefficients, while the stars indicate the adjusted significance levels. The factor Year compares guild composition between year 2002 and year 2001, while Management compares primary sites and managed sites. $p < 0.05$ (*), $p < 0.01$ (**), $p < 0.001$ (***).

| Grazer | Parasitoids | Plant-Suckers | Xylophages | Mycetophages | Plant-Chewers | Saprophages | Zoophages |
|---|---|---|---|---|---|---|---|
| 0.15 *** | 1.38 ** | 0.57 *** | 2.21 *** | 1.69 *** | 1.97 *** | 3.04 *** | 1.63 *** |
| 2.75 *** | 1.07 ns | 0.91 ns | 1.07 ns | 0.71 ns | 0.96 ns | 0.66 ns | 0.60 *** |
| | 0.99 * | | | | | 1.03 ** | 1.01 * |

## 4. Discussion

### 4.1. Composition of Canopy Communities

With 1152 fogging samples, this is the largest data set of arboreal arthropod communities collected in Central Europe published so far. Tree-specific data that cover all taxa and most of the ectophytic arthropod fauna can only be obtained by insecticidal knock-down. Fogging thus makes it possible to analyse the distribution of species reliably, and irrespectively of whether they are juveniles or adults, flying or flightless [27]. The low within-variability between tree genera in different sites is impressive evidence of this.

The data provided the most complete picture of tree-specific taxa composition to date, and showed that the distribution of the arthropod taxa follows regular patterns in the canopy (Figures 2 and 3). Due to high spatio–temporal variability, these patterns emerged only on a very large dataset. Eight 'major' taxa, collected from each tree in greater numbers, together accounted for more than 80% of all the arthropods in the trees (Figure 1, Table S2), and formed the basic structure of any community. Ordered by their frequency, these were Diptera, Homoptera, Hymenoptera, Heteroptera, Coleoptera, Psocoptera and, also, in lower frequencies, Lepidoptera and Araneae. In terms of numbers, Diptera dominated all trees, followed by Homoptera, which occurred predominantly on deciduous trees, and mainly as juveniles, while Psocoptera were collected preferentially from conifers. The data confirmed previous findings that broad-leaved and coniferous trees support

different communities [13]. Contrary to what is often assumed [35], Lepidoptera are neither numerically nor functionally dominant in trees (Figures 3B and S3), although they were regularly collected by fogging, with, on average, 100 and 134 specimens from coniferous and deciduous trees, respectively. This assessment is probably due to the mass occurrences and economic damage that caterpillar outbreaks cause, as this was also recorded in our data from the Steigerwald, 1996. Although fogging probably underestimates the caterpillar abundance of many leaf-feeding microlepidopterans, it is unlikely to change the overall picture of community composition. Spiders regularly occur in higher abundance in the trees, and form communities that may also distinguish tree species [36]. In temperate forests, at least 20% of the spider biomass is found in the canopy and understory [37], suggesting that their ecological importance as generalist predators remains underestimated. Spider abundance may be limited by bottom-up processes, via resources and abiotic factors, rather than top-down processes [15]. Beside the 'major' taxa, less common 'minor' taxa and other rare groups complement the canopy communities. Some taxa like Ephemeroptera and Plecoptera may reach extremely high abundance for a short time. After mass hatching, they can provide an enormous short-term pulse of biomass to the ecosystem [38].

Acarina, Collembola and Thysanoptera are also of great functional importance, but were not considered in this work. As mentioned above, their abundance distribution varied so much among trees that it did not allow for reliable analyses, but rather indicated the methodological shortcomings. For example, Acarina and Collembola were collected with less than 15 specimens on almost half of all fogged trees, whilst more than 10% of all sampled trees had no mites or springtails at all. The maximum number of mites was 4800 specimens on one tree. Collembola were collected in lower numbers overall, but their numbers also varied from 0 to 3000 per tree. Thysanoptera showed the same distribution. These methodological shortcomings apply to all endophytic groups which develop in deadwood, or which live in bark crevices. A complete monitoring of the arboreal biodiversity, including microarthropods, sedentary and endophytic groups, has not yet been carried out, which makes taxonomic and faunal analyses and the functional characterization incomplete and prone to error. However, this paper provides an overview of the general functional importance of arboreal canopy arthropods, and the need for their consideration in ecosystem analyses. Due to advances in DNA barcoding technology, a complete recording is only a matter of time; Diptera are a good example of this [39,40].

### 4.2. Guild Composition

The focus of previous guild studies of canopy arthropods was on species composition [12,41]. However, the functions that canopy arthropod communities perform in ecosystems, and the services they provide, are based on the total number of specimens, not the number of species. Here, we provide a first overview of the functional importance of complete arboreal arthropod communities. Even individual species, like the predacious ant *Formica polyctena*, can decisively influence and determine biodiversity and ecosystem function [7,42]. Little is known about the factors that determine guild composition in the canopy, but there is no evidence of guild constancy, as shown by comparisons of forests between and within countries.

Guild composition between conifers and deciduous trees was surprisingly similar, differing only in the proportion of grazers confirming earlier observations [13]. Few differences in guild composition were significant, due to variance in taxa abundance distributions between individual tree genera and forests. This points toward the great importance of spatio–temporal factors such as year of study, weather conditions, forest type, etc., in determining community composition [6], and illustrates how difficult it is to interpret studies from different areas and years. To examine how communities in primary forest remnants differed from those in managed study areas, we focused on oaks in the Bialowieza Forest, to achieve the highest possible homogeneity. We found significant differences for grazers, which occurred in larger proportions in primary sites, and zoophages, which were collected in higher proportion in commercial forests (Table 1). The other guilds did not

differ from each other in their relative proportions. As the studied sites were located in the same forest matrix, this indicates that the forest matrix reduces the differences between sites, but it also shows how much forest management affects guild composition.

According to our results, tree-specific properties, such as leaf cover, height or girth in breast height, have only a small influence on guild composition. In contrast, the main drivers of guild composition are tree genera, year, and the study area, which highlight site-specific spatio–temporal differences. The significance of the binary factor Management on all spatial scales was particularly surprising, and was likely caused by changes in plant and structural diversity, and the microclimatic conditions [19]. This also suggests that, with the loss of primeval forests in our Central European cultural landscape, the evaluation basis for assessing the impact of forest management has also been lost. It remains to be seen to what extent these results will be relevant to forestry. However, recognizing the importance of the canopy fauna for maintaining background functions and resilience in forest ecosystems is a prerequisite for more sustainable management, especially given the impending changes in climate change [6,43].

### 4.3. Functional Importance of Canopy Arthropods

Although insects account for little biomass, they are essential to the functioning of forests and ecosystem services [7]; however, biomass says little about function. Assessing the functional relevance of arthropods requires the assignment of all specimens to their feeding guild, which can then serve as a surrogate for function. Their high abundance in the trees alone suggests that canopy arthropods are major players in forest ecosystems. There is also low faunistic and functional overlap of the canopy with the soil habitat [44], which indicates the unique importance of the canopy fauna. Furthermore, recent research has shown that the canopy fauna significantly affects soil-inhabiting assemblages and the interaction networks of ectomycorrhizal fungi, saprophytes and pathogens [45,46].

Arthropods are not only food to a vast range of animals but also determine the reproductive success of various forest tree species, e.g., pollinators such as *Acer*, *Salix*, *Tilia*, Rosaceae and herbaceous plants. As herbivores and detritivores, they link plant biomass to higher trophic levels, and recycle nutrients [47]. The fogging data confirmed their dominance in herbivore–tree interactions in canopy communities, where herbivores can represent between 40% and 50% of all specimens [9,12]. The importance of canopy herbivores becomes particularly evident when they occur in high abundance, and cause economic damage [48]. Examples can be found among leaf-feeding caterpillars (Lepidoptera), plant wasps (Symphyta, Hymenoptera), sap suckers (Homoptera) or wood-boring saproxylic insects such as Scolytinae (Curculionidae) or Buprestidae [49]. Most pest species are under natural biological control by predators, parasitoids or pathogens [47,50] of which many live in the canopy, but whose contribution to the suppression of pest species populations is still little understood. However, the actual impact of herbivores in trees is questionable. Investigations in temperate forests in South Africa reported the dominance of zoophages [4], and similar results were found for *Fraxinus* trees in floodplain forests in Germany [6]. To date, however, too few fogging studies have analysed the entire fauna, to draw more general conclusions.

## 5. Conclusions

The fogging data provided knowledge on the taxa composition of tree-specific arthropod communities, which was an important step toward understanding their functional importance in forest ecosystems. Even though the autecology of many arboreal species is known, the ecological significance of the whole canopy community remains poorly understood. This could not only prove detrimental to the sustainable use of forests, but might become a major disadvantage under climate change, which will greatly affect forest diversity, stability and functionality [6].

**Supplementary Materials:** The following supporting information can be downloaded at: https://www.mdpi.com/article/10.3390/d14080660/s1, Figure S1: Ranked taxa abundance in the canopy communities of all fogged tree genera total number of individuals in brackets below taxon rank. The darker the shade of green, the higher the rank of the particular group; the highest values are shown in bright red. Tree genera on the y-axis number of foggings in brackets. Included are all tree genera fogged in more than five trees. Mass occurrence of individual taxa (see text) are excluded. Figure S2: Pairwise Wilcox tests with Benjamini-Hochberg p-value adjustment of eight feeding guilds on all tree genera. Only tree genera with more than ten trees fogged are considered. Visualised are *p*-values; numbers are mean differences in percentages of specimens per tree and feeding guild percent individuals on y-axis minus percent individuals on x-axis. Non-significant comparisons not shown. Figure S3: Horizontal bar-plots show multiple testing-adjusted significance levels of the mean abundance of taxa between coniferous and deciduous trees (Wilcox tests). Dotted red lines delineate significance levels on the log x-axis ($p = 0.05$; $p = 0.01$ and $p = 0.001$). Mean numbers of Arachnida, Aphids, Coleoptera, Blattodea, Lepidoptera (mainly caterpillars), Neuroptera and parasitic Hymenoptera differed not significantly between coniferous and deciduous trees. Number of foggings considered was 902. Only trees fogged more than 5 times were considered, mass events were excluded. Table S1: Distribution of the 1152 foggings carried out in different sites in Central Europe with forest type specific information. Research was carried out from 1995 to 2020; unm = unmanaged forests, com = commercial forest, prim = primary forest). Table S2: Number of arthropods per taxon sorted by individual numbers. The total number of trees fogged was 1152. Because of their high local abundance and ecological impact aphids and ants are shown separately. Thysanoptera, Acarina, and Collembola were not included due to large differences in numbers between sampled trees. Table S3: Assignment of arthropods to feeding guilds as they were collected by fogging in June. Only few individuals of adult Lepidoptera were collected in June, which were not included in the analysis. Taxa were also assigned to one of three categories according to their abundance and constancy in the canopy: major = regularly frequent, minor = regularly rare, sporadic = sporadic occurrence. Table S4: Full model results of the regression models of guild composition between tree genera shown in MS.

**Author Contributions:** Conceptualization, A.F.; Data curation, A.F.; Formal analysis, A.F. and T.M.; Funding acquisition, A.F. and K.E.L.; Investigation, A.F.; Resources, K.E.L.; Writing—original draft, A.F. and T.M.; Writing—review & editing, K.E.L. All authors have read and agreed to the published version of the manuscript.

**Funding:** This work summarises many research studies that have been carried out during the last 25 years. Financial support from DFG, the VW-Foundation and the Department of Zoology and Tropical Biology is acknowledged. AF was partially supported by the BAYERISCHE Forschungsstiftung, grant number AZ-1365-18 FORTiTher, TP-4.

**Institutional Review Board Statement:** Not applicable.

**Data Availability Statement:** Not applicable.

**Acknowledgments:** We would like to thank the many governmental authorities which granted permission to carry out the fogging experiments in all countries. We also thank the many students for their help in field and in the lab. Without this support, this huge work could not have been done. Identification of taxa were made by many specialists: Diptera: W. Schacht †, A. Stark, H.-P. Tschorsnig; Heteroptera: A. Melber, M. Gogala, F. Schmolke; Plecoptera: H. Reusch †, K. Enting; Formicidae, Ichneumonidae and other parasitic Hymenoptera groups: K. Horstmann †; Aculeate Hymenoptera: M. Kraus; Formicidae: B. Seifert; Araneae: H. Stumpf, S. Otto; Coleoptera: B. Büche, P. Sprick; Phytophages, mainly Chrysomelidae and Curculionidae: J. Esser; Cryptophagidae and other families: D. Siede † Ptiliidae, A. Lompe; *Trixagus* and other families: K. Renner; *Epuraea pallescens*: B. Feldmann; Staphylinidae: H. Fuchs; *Mordellistena purpureonigrans*: H. Meybohm; *Bibloplectus*: V. Brachat; Pselaphinae: A. Kopetz; Cantharidae: W. Rücker. Voucher specimens are deposited in the collection of A. Floren. The research was decisively supported by the institute workshop, particularly by Norbert Schneider, who prepared much of the equipment used in the field.

**Conflicts of Interest:** The authors declare that the research was conducted in the absence of any commercial or financial relationships that could be construed as a potential conflict of interest.

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
