# Peer review of "Diversity and Functional Relevance of Canopy Arthropods in Central Europe"

_diversity, doi:10.3390/d14080660_

Round 1

Reviewer 1 Report

In their study, Floren et al. examined the taxonomic structure and feeding guild composition of canopy arthropods in Central Europe. Based on an impressive dataset (> 3.2 Mio arthropod individuals), the authors found that the taxonomic structure is widely consistent among tree genera. The authors found further that guild composition differed only slightly between deciduous and coniferous trees as well as among tree genera.

I really like the study and I am convinced that it will make a very nice contribution to the existing literature. However, there are several issues which await clarification and revision. Especially the way how the statistical analyses are described and structured is hard to follow and should be carefully revised.

Although the manuscript reads overall well and is (apart from the abovementioned point), easy to follow, there are some issues here and there. Please use the revision as well to check the language, especially the punctuation.

Major comment

The description/ structure on how you conducted the statistical analyses is confusing and should be carefully revised to improve clarity. Many aspects become only clear once the reader knows the results. Most importantly, it must be clearly separated which analyses were conducted across all countries and sampling sites and which analyses were conducted only on Polish Quercus trees. Some (but not all) points:

l. 148: This is analysed in detail for Quercus: Difficult to understand in context with the sentence before. How can you analyse the importance of tree genera based on one tree genus?

ll. 149/150, e.g. ..excluded mountain forests in Romania and Slovania: this is confusing as you write a sentence before that you focus on Quercus trees in Poland for this analysis.

l. 156 please, specify exactly for what exactly you tested for.

l. 163 ff. Here I am quite lost: what was investigated based on which data. It’s hard to distinguish among analyses restricted to oak trees and those which were analysed across site

Maybe consider subheadings to structure the analyses in an unambiguously way.

 Minor comments

l. 37 specify that Saxony-Anhalt is in Germany to provide better orientation for the reader

l. 51 cite the study by Erwin

l. 67 …leaf morphology, crown structure AND phytochemistry, which is certainly of high importance for all the herbivores

l. 79 the sentence ends abruptly … it’s not possible to evaluate the last part of your introduction. Please ensure that ALL text parts are included in a revised version of this MS.

l. 119/120 “…, which in some cases …” the last part of this sentence is a bit confusing. As you directly explain in the following sentences that some orders were split more precisely, you could basically delete this part.

l. 127 ontogenetic shifts: Based on S-Table 3, I assume you excluded all adult Lepidoptera? If so, please state it clearly. Furthermore, it would be helpful to specify in S-Table 3 precisely which stages you considered for holometabolous insects  

l. 130 integrating over several month… what exactly do you mean, please clarify (maybe related to my main comment on seasonality before)

l.  138 what was the percentage of Formica polyctena across the ant samples? Please provide this information.

l. 146 overall distribution: do you mean “taxonomic composition within individual tree species” or “overall arthropod abundance”?

l. 149 you state that the guild analyses base on samplings restricted to June. How does it look for the general sampling and other analyses? Could it have influenced your comparison among trees?

l. 151 how you evaluated “rare” and “exceptionally high”? Was there an objective threshold?

l. 168 sorry if I missed it, but what indicates the ‘area’ variable here?

l. 163 …six relative proportions.. what do you mean? It should be clear without referring to sections within the Results part

L 166 & 174 I assume it’s a PERMANOVA, if so state it clearly

l. 167 cite vegan package here

l. 211 Why Araneae in the text and Arachnida in the Figure 3A? Same for Aphidoidea and Aphidae?

l. 211ff. why is Aphidoidea considered a “minor taxon” although it’s ranked before Arachnida (major taxon) in Figure 3A?

229ff so these “exceptional” events of mass appearances were excluded from analyses? If so, this part should be transferred to the Methods.

l. 251 & Fig. 3B Just out of interest: Given the fact that most caterpillars feeding on trees are associated with deciduous trees, why the pattern tends here towards conifers? Any explanation? Could it result from the fact that most shelter building caterpillars (leaf rollers, folders etc. = on deciduous trees) are underrepresented in fogging samples? Maybe this could add 1-2 sentences to the discussion, where you discuss potential effects on the “fogging”-method on guild/ Taxon composition

l. 268 indicate p value thresholds of individual significance levels in the legend

l. 285-287 Please, describe in the legend what the values and the colours of individual cells indicate and how the Figure should be interpreted. Together with its legend, the Figure should be self-explaining.

S-Figure 1 Please, indicate here clearly the meaning of colours and numbers (which are not in brackets)

l. 288 here an throughout the manuscript: check how you refer to the test (see comment on PERMANOVA) and cite correctly in brackets  

l. 358 See my comment above: What are the implications of fogging? I would assume that you miss a large fraction of “Microlepidopteran” larvae as they form various types of leave shelters. This “guild” of shelter builders can make up substantial fractions in temperate forests. I am not questioning the overall pattern based on this amazing and certainly robust dataset. But I think the methodological shortcomings of fogging and how they could have influenced your findings should be discussed. DNA barcoding (l. 373) will certainly facilitate this method in the future but it will also not solve the problem of missing taxa (e.g., internal feeders, shelter builders, etc.) in the sample.

Furthermore: How could have the timing of the sampling have influenced the taxonomic ranking you found? See also my previous comment to clarify the seasonal aspect: WHEN did you conduct the fogging (whole season or ALWAYS in June?).

l. 392 where is Figure 6?

l. 421 reference in brackets (check throughout MS)

l. 431 Africa

Furthemore: Check carefully the references, e.g. l. 515: Title is missing…

Author Response

Dear editors,

Thank you for the positive and valuable criticism, which significantly improved the MS!

All comments are addressed in detail by our point by point answer.

In accordance with the comments of reviewer 1, we have revised and restructured the methodological part concerning the statistical analysis in particular, so that the ambiguities have hopefully now been eliminated.

We have also addressed in detail the important criticism of reviewer 2 by explaining why Diptera were not included in the analyses and why Acarina, Collembola, and Thysanoptera were not addressed due to methodological uncertainties.

We have included all minor comments and thank the reviewers for their accuracy in reading and commenting on the MS.

We hope that the MS can be accepted

Scincerely

for the authors

Andreas Floren

Reviewer 2 Report

General

The authors present an impressive analysis of an enormous dataset. This is by far the most comprehensive study of its kind and will contribute a lot to the knowledge of canopy arthropods. I am not able to find many flaws but I have a number of questions that need to be answered.

Why not use the individual feeding guild on the species level when the species is known. E.g. Diptera? You (and other experts) have spent so much time identifying so why use only the higher taxa, which will be less precise. You say on L144-145 that Diptera could not be assigned to feeding guilds. I read that it means that they are taken out from the analyses, like Thysanoptera, Collembola and mites. If so, this reduction must have an impact on the overall results as, correctly said, Diptera is by far the most abundant (and also species-rich) taxon in the canopies. I also see a non-convincing argument to take out Collembola and mites (see next paragraph).

L124-125 and L197-199: Great differences between trees are very common and how can that be a valid reason to exclude them? Those would probably contribute to understand more on the landscape level (management) and impact of the tree-related factors themselves (height, age, dbh etc.). Within especially the mites you would have a number of guilds. To a lesser extent the same for Collembola. But then, you would of course need them identified. I do not see a good reason to exclude those three taxa (and Diptera if I understand correctly). Lack of taxonomic expertise, probably huge numbers or a time consuming identification process would have been perfectly valid reasons, and this includes for Diptera too. Please explain in more detail.

Following on L199, there are many groups that cannot be sampled representatively; those curculionids locking their extremities upon disturbance, those minute wasps that get stuck on the leaves because there is not a free fall all the way to the ground or because you did the sampling in the early morning where there is more moisture on the foliage due to the drop in temperature just before sunbreak. Those trapped in crevices.

You claim that eight taxa are universally dominating. Yes, but what about Acarina and Collembola as you left them out? It should be discussed. Thysanoptera probably not that much. In the discussion I also miss the answer to why are those eight dominating? Is the answer as simple as ‘trees are trees’? Please add a paragraph.

L46-47: All specimens? I doubt that for new species and lesser known taxa. You must make some assumptions. It is correct that studies for the most restrict analyses to fewer taxa (L57), simply because their feeding guild is not known on the species or even genus level.

L110-111: Did you consider ‘summer dormancy’? Arthropod activity usually follows a bimodal pattern going down in the warmest and driest period. Also, “before juveniles had pupated” assumes that they all follow a similar life cycle, which is not the case.   

Figure 4: What are the numbers? I also guess that the red blocks are the ones being dominant. Please explain in the legend. Also, the choice of colors does not match between A and B. Is there a reason for that?

Explaining the Adonis results on page 8 is fine but I miss a concluding remark in that paragraph telling the reader what this really means. This will then make a nice connection to the next paragraph.

L377-379: This is a strong argument and should be backed up by literature. One single species can provide crucial services. Biodiversity itself is such a service and should not be made redundant.

Minor issues

L11-12: font is different

L13: maintain

L16-17 & L45-46: Fogging…. Explain “same number as they occur in the trees”. Did you mean “same proportion”?

L27: Within-group

L79: Something is missing

L92: Last word should be ‘are’

L119-120: Unclear, please explain

L167: Add [ ] at reference 33

L178: Inconsequent use of parentheses style compared with L179 and L180

L184-192: You are saying the same twice. Pick one version

L230: Lowercase on the species author

L248: Comma placed wrongly, should be 192,516

L250: Change but with while

L252: This paragraph does not support the differences between conifers AND deciduous trees. Change ‘differences’ with ‘among’ maybe?

L288: “We used…”

Figure 5: What does Gattung mean? Lost in translation maybe?

L322: Redundant space after 218,512

L326-328: The beginning of this sentence is strange

L337: “Management” or “Primary” as in the table?

Table 1: Typo in the heading, should be “Plant-suckers”

L350: Figs, not Fig’s

L358: Add [ ] at reference 35

L421: Add [ ] at reference 46

References:

11: Wint, not WINT

26: Missing the title?

27: Missing the title

33: Wrong style

Author Response

(The authors gave the same response as above.)

Round 2

Reviewer 1 Report

The authors clarified most of the concerns I had on the previous version. I found only some minor issues, which should be addressed. Especially the methods section could still be slightly improved.

General:

I still think that the authors should again carefully check the spelling. (For example: Some of the sentences are rather long and could be split. Many commas are missing, which hampers understanding.)

Specific points:

l. 15 please state here clearly what you mean with „taxa“

ll. 34-38: I do not see what the “number of red-listed beetle species” (ll. 36-38) tells us about our “knowledge of canopy arthropods” (sentence before, l. 34-36). The number of red-listed spp. simply indicates that a lot of these canopy-dwelling taxa are treated, but by no means whether they are well studied or not.

l. 48 “study” instead of “MS”

l. 67 change to: “…leaf morphology, crown structure, and…

l. 70 please specify clearly (here and throughout the study) that “area” means “study area” or “sampling site” – otherwise the term leads to confusion

l. 80 maybe “evaluate” works better instead of “calibrate”?

l. 82 “25 years” maybe specify here directly in brackets the exact time period: (1995-2020)

l. 120 “Nevertheless,…” This sentence is redundant to the one before and thus could be deleted

l. 123 “all arthropods were sorted to major taxa” – please specify here clearly the level (order, suborder, family). I strongly recommend to avoid the term “major taxa” here, as you use the same term in a completely different context within your results section (i.e., when you describe how arthropod groups are arranged: major vs. minor taxa). This is confusing.

ll. 152/153 “Diptera…” this sentence is entirely redundant to the statement made in ll. 138-141 and should thus be deleted.

ll. 159-161 It is still a bit confusing when you write here that you focussed on oaks, but then describe “Stratum 1” and “Stratum 2” below (which do include ALL tree species and not only oaks) – you only focus on oaks in “Strata 3 and 4”. This should be clarified.

l. 168 either (N=319) or leave the brackets out

ll.162-276 Just a suggestion: Maybe you could think about another term instead of “Stratum” as this has a very specific meaning in vegetation (forest) ecology. “Level” for example??

l. 180 “Exceptionally…” I do not fully understand the logic behind this sentence. Please clarify.

ll. 191 ff. It is not necessary to describe here what the violin plots indicate. That’s more important in the respective Figure legend (and there it is already written).

l. 202 “adonis2” is an R function implemented in the vegan package and NOT a statistical method (as you correctly mention in ll. 209ff.). Please correct this here as well.

l. 226 replace “as implemented in” with “using”

ll. 240ff I am still wondering whether there was any objective criterion to distinguish among “major taxa” and “minor taxa”?? As it is written the demarcation sounds a bit arbitrarily, i.e., what is “regularly” and what are “high numbers”? Please specify this somewhere.

Fig. 3A: split y label into two words

Fig 3C: it should be: “p” 0.001

l. 324 “relative proportions” where is this coming from?? If this is part of the Figure it must be clarified: relative prop. of what?

ll. 412/413 “caterpillar gradations” do you mean “caterpillar outbreaks”?

ll. 413-416 although I understand what you aim to say, this sentence must be rephrased for clarity

S-Table 1 Please specify in the Table legend what “com”, “unm”, and “prim” indicate

Author Response

To the reviewer:

Thanks for the comments which had all been considered in the revised version (see below). We hope this work highlights the importance of arboreal arthropods to forest ecology! 

For the authors

Andreas Floren

Open Review

English language and style

( ) Extensive editing of English language and style required
( ) Moderate English changes required
(x) English language and style are fine/minor spell check required
( ) I don't feel qualified to judge about the English language and style

Yes

Can be improved

Must be improved

Not applicable

Does the introduction provide sufficient background and include all relevant references?

(x)

( )

( )

( )

Are all the cited references relevant to the research?

(x)

( )

( )

( )

Is the research design appropriate?

(x)

( )

( )

( )

Are the methods adequately described?

( )

(x)

( )

( )

Are the results clearly presented?

(x)

( )

( )

( )

Are the conclusions supported by the results?

(x)

( )

( )

( )

Comments and Suggestions for Authors

The authors clarified most of the concerns I had on the previous version. I found only some minor issues, which should be addressed. Especially the methods section could still be slightly improved.

General:

I still think that the authors should again carefully check the spelling. (For example: Some of the sentences are rather long and could be split. Many commas are missing, which hampers understanding.)     the MS was checked again as noted.

Specific points:

  1. 15 please state here clearly what you mean with „taxa“
  2. 34-38: I do not see what the “number of red-listed beetle species” (ll. 36-38) tells us about our “knowledge of canopy arthropods” (sentence before, l. 34-36). The number of red-listed spp. simply indicates that a lot of these canopy-dwelling taxa are treated, but by no means whether they are well studied or not.

The finding of so many Red List species indicates how inadequately canopies are still studied and how little is known about the frequency distribution of species and their functional relevance.   

  1. 48 “study” instead of “MS” corrected
  2. 67 change to: “…leaf morphology, crown structure, and… corrected
  3. 70 please specify clearly (here and throughout the study) that “area” means “study area” or “sampling site” – otherwise the term leads to confusion corrected
  4. 80 maybe “evaluate” works better instead of “calibrate”? corrected
  5. 82 “25 years” maybe specify here directly in brackets the exact time period: (1995-2020) corrected
  6. 120 “Nevertheless,…” This sentence is redundant to the one before and thus could be deleted corrected
  7. 123 “all arthropods were sorted to major taxa” – please specify here clearly the level (order, suborder, family). I strongly recommend to avoid the term “major taxa” here, as you use the same term in a completely different context within your results section (i.e., when you describe how arthropod groups are arranged: major vs. minor taxa). This is confusing. corrected
  8. 152/153 “Diptera…” this sentence is entirely redundant to the statement made in ll. 138-141 and should thus be deleted. corrected
  9. 159-161 It is still a bit confusing when you write here that you focussed on oaks, but then describe “Stratum 1” and “Stratum 2” below (which do include ALL tree species and not only oaks) – you only focus on oaks in “Strata 3 and 4”. This should be clarified.

The sentences referring to the oak analysis have been moved to the the correct position that refers to the analysis of the oaks.

  1. 168 either (N=319) or leave the brackets out corrected

ll.162-276 Just a suggestion: Maybe you could think about another term instead of “Stratum” as this has a very specific meaning in vegetation (forest) ecology. “Level” for example??

rephrased

  1. 180 “Exceptionally…” I do not fully understand the logic behind this sentence. Please clarify. 194 Sentence reformulated
  2. 191 ff. It is not necessary to describe here what the violin plots indicate. That’s more important in the respective Figure legend (and there it is already written). deleted
  3. 202 “adonis2” is an R function implemented in the vegan package and NOT a statistical method (as you correctly mention in ll. 209ff.). Please correct this here as well. corrected
  4. 226 replace “as implemented in” with “using” . corrected
  5. 240ff I am still wondering whether there was any objective criterion to distinguish among “major taxa” and “minor taxa”?? As it is written the demarcation sounds a bit arbitrarily, i.e., what is “regularly” and what are “high numbers”? Please specify this somewhere.

The "major taxa" include all groups that represented on average minimum 5% of all individuals in canopy communities. Exceptions were Lepidoptera (3%) and Arachnida (2%), which were considered “major taxa” because they were also collected from every tree.     Information added (l. 274)

The second group are the “minor taxa”, which occurred on most trees but in significantly lower numbers providing on average less than 1% of all specimens to a community. Information added (l. 284)

 Fig. 3A: split y label into two words    corrected

Fig 3C: it should be: “p” ≤ 0.001      corrected

  1. 324 “relative proportions” where is this coming from?? If this is part of the Figure it must be clarified: relative prop. of what? clarified
  2. 412/413 “caterpillar gradations” do you mean “caterpillar outbreaks”?
  3. 413-416 although I understand what you aim to say, this sentence must be rephrased for clarity rephrased

S-Table 1 Please specify in the Table legend what “com”, “unm”, and “prim” indicate   corrected

Submission Date

14 July 2022

Date of this review

10 Aug 2022 13:23:35